# Use of Inhaled Corticosteroids and Risk of Acquiring *Haemophilus influenzae* in Patients with Chronic Obstructive Pulmonary Disease

**DOI:** 10.3390/jcm11123539

**Published:** 2022-06-20

**Authors:** Raza Ul Mohsin, Christian Kjer Heerfordt, Josefin Eklöf, Pradeesh Sivapalan, Mohamad Isam Saeed, Truls Sylvan Ingebrigtsen, Susanne Dam Nielsen, Zitta Barrella Harboe, Kasper Karmark Iversen, Jette Bangsborg, Jens Otto Jarløv, Jonas Bredtoft Boel, Christian Østergaard Andersen, Henrik Pierre Calum, Ram B. Dessau, Jens-Ulrik Stæhr Jensen

**Affiliations:** 1Department of Internal Medicine, Section of Respiratory Medicine, Herlev and Gentofte Hospital, University of Copenhagen, 2900 Hellerup, Denmark; christian.kjer.heerfordt@regionh.dk (C.K.H.); josefin.viktoria.ekloef@regionh.dk (J.E.); pradeesh.sivapalan.02@regionh.dk (P.S.); mohamad.isam.saeed.02@regionh.dk (M.I.S.); truls.sylvan.ingebrigtsen@regionh.dk (T.S.I.); jens.ulrik.jensen@regionh.dk (J.-U.S.J.); 2PERSIMUNE & CHIP, Department of Infectious Diseases, Rigshospitalet, University of Copenhagen, 2100 Copenhagen, Denmark; susanne.dam.poulsen@regionh.dk; 3Department of Pulmonary Medicine and Infectious Diseases, Copenhagen University Hospital, 3400 North Zealand, Denmark; zitta.barrella.harboe@regionh.dk; 4Department of Clinical Medicine, Faculty of Health and Medical Science, University of Copenhagen, 2100 Copenhagen, Denmark; 5Department of Internal Medicine, Section of Cardiology, Herlev and Gentofte Hospital, University of Copenhagen, 2730 Herlev, Denmark; kasper.karmark.iversen@regionh.dk; 6Department of Clinical Microbiology, Herlev and Gentofte Hospital, University of Copenhagen, 2730 Herlev, Denmark; jette.bangsborg@regionh.dk (J.B.); jens.otto.jarloev@regionh.dk (J.O.J.); jonas.bredtoft.boel.01@regionh.dk (J.B.B.); 7Department of Clinical Microbiology, Amager og Hvidovre Hospital, University of Copenhagen, 2650 Hvidovre, Denmark; christian.oestergaard.andersen@regionh.dk (C.Ø.A.); henrik.pierre.calum@regionh.dk (H.P.C.); 8Department of Clinical Microbiology, Slagelse Hospital, 4200 Slagelse, Denmark; ramd@regionsjaelland.dk

**Keywords:** chronic obstructive pulmonary disease, haemophilus influenzae, inhaled corticosteroids

## Abstract

**Background:** Inhaled corticosteroids (ICS) are widely used in chronic obstructive pulmonary disease (COPD), despite the known risk of severe adverse effects including pulmonary infections. **Research Question:** Our study investigates the risk of acquiring a positive *Haemophilus influenzae* airway culture with use of ICS in outpatients with COPD. **Study Design and Methods:** We conducted an epidemiological cohort study using data from 1 January 2010 to 19 February 2018, including 21,218 outpatients with COPD in Denmark. ICS use 365 days prior to cohort entry was categorised into low, moderate, and high, based on cumulated ICS dose extracted from a national registry on reimbursed prescriptions. A Cox proportional hazards regression model was used to assess the future risk of acquiring *H. Influenzae* within 365 days from cohort entry, and sensitivity analyses were performed using propensity score matched models. **Results:** In total, 801 (3.8%) patients acquired *H. Influenzae* during follow-up. Use of ICS was associated with a dose-dependent increased risk of acquiring *H. Influenzae* with hazard ratio (HR) 1.2 (95% confidence interval (CI) 0.9–1.5, *p* value = 0.1) for low-dose ICS; HR 1.7 (95% CI 1.3–2.1, *p* value < 0.0001) for moderate dose; and HR 1.9 (95% CI 1.5–2.4, *p* value < 0.0001) for high-dose ICS compared to no ICS use. Results were confirmed in the propensity-matched model using the same categories. **Conclusions:** ICS use in outpatients with COPD was associated with a dose-dependent increase in risk of isolating *H. Influenzae*. This observation supports that high dose ICS should be used with caution.

## 1. Introduction

Inhaled corticosteroids (ICS) are widely prescribed to decrease the risk of acute exacerbations in patients with chronic obstructive pulmonary disease (COPD) and are currently recommended to those with eosinophilic inflammation (i.e., blood eosinophils >300 cells/μL) or asthma–COPD overlap [1,2,3]. However, ICS use may increase the risk of harmful side effects, including pneumonia [4,5,6], especially in population groups susceptible to ICS-induced pneumonic infections such as older patients with low body mass index (BMI) and severe airflow limitations, particularly when receiving high ICS doses [1]. It is true that ETHOS and IMPACT show a reduction in all-cause mortality with ICS use, though this conclusion is considered inaccurate by some [7,8]. Moreover, a network metanalysis of ETHOS, KRONOS, IMPACT, and TRILOGY studies also show an increased risk of pneumonia with no improvement in mortality despite decreased risk of acute exacerbation of COPD with use of ICS in dual- or triple-fixed dose containers [8]. Eventually, some other studies have also shown none [9,10] or minimal [11] improvement in survival benefits associated with ICS use in COPD.

*Haemophilus Influenzae* is frequently cultured from the lower airways of COPD patients in stable periods, which has been linked to worsening of daily symptoms [12]. It is also a major cause of acute exacerbations, and these can accelerate disease progression [13]. Nevertheless, thus far, no clinical studies have systematically explored the influence of different dose regimens of ICS on the risk of isolating *H. influenzae* in COPD patients. The aim of our study was to determine the risk of acquiring a positive *H. influenzae* airway culture after use of different accumulated doses of ICS by COPD outpatients. 

## 2. Methods

### 2.1. Data Sources 

In our study, we used:IThe Danish Register of Chronic Obstructive Pulmonary Disease (DrCOPD), which is a national register established in 2008. It contains information about the quality of treatment of COPD patients from all over Denmark. Data are extracted from inpatient and outpatient pulmonary clinic visits. The following variables were used: age, forced expiratory volume in 1 s (FEV_1_), smoking status (i.e., active, former, never), BMI, dyspnoea registered using the Medical Research Council (MRC) Dyspnoea Scale [14].IIThe Danish National Patient Registry (DNPR), which is used to determine the comorbidities in the study population. It contains records from all Danish hospitals since 1977 and all hospital outpatient visits since 1995 [15].IIIThe Danish National Database of Reimbursed Prescriptions (DNDRP), which is used to assess exposure to ICS. It contains records of all prescribed and reimbursed medications obtained in the Danish community and hospital-based pharmacies since 2004. Drugs are coded according to Anatomical Therapeutic Chemical (ATC) classification [16].IVMicrobiological data, which were obtained from the Clinical Microbiology Departments in Eastern Denmark (i.e., Zealand and Capital Regions) to retrieve *H. influenzae* culture results.

All these registers contain an encrypted version of a ten-digit civil personal registry number, unique to every patient, which allows for anonymized linkage and all-covering follow-up [17]. 

### 2.2. Study Population

This study included COPD outpatients aged 30 years and older, registered from 1 January 2010 to 19 February 2018 in the DrCOPD (Figure 1). Patients with only in-hospital visits were excluded, as these hospital contacts lack registration of essential clinical information (i.e., BMI, smoking status, severity of airflow obstruction, and COPD Assessment Test (CAT) score). Patients from the western part of Denmark were also excluded due to the lack of access to microbiological data from this region.

We excluded patients from whom *H. influenzae* was cultured from lower respiratory tract samples within the last 12 months prior to cohort entry, patients who were on treatment with any disease-modifying anti-rheumatics drugs (Anatomical Therapeutic Chemical (ATC) codes: L04AX03, L01AA01, A07EC01, L04AD01, L04AA13, L04AX01, L04AA06, P01BA02) 12 months prior to cohort entry, and patients who were diagnosed with either immunodeficiency (International Classification of Disease (ICD-10) codes: D80–84, D86, D89) or malignant neoplasm (ICD-10 codes: C00–C97) within 5 years prior to inclusion to the cohort. Appendix A lists the ICD-10 codes used to define comorbidities in the study population. Patients were followed for 365 days from cohort entry (i.e., first registry in DrCOPD) until the first H. influenzae-positive sample, death, or end of study period on 19 February 2018. 

### 2.3. Exposure to ICS 

Exposure to ICS was quantified by calculating the accumulated dose using all ICS prescriptions reimbursed within 365 days prior to study entry. This was then divided into tertiles of low, moderate, and high ICS dose users. Non-users were used as a reference group. The ICS types identified were budesonide, beclomethasone, fluticasone, mometasone and ciclesonide. All of these were converted to budesonide-equivalent doses, with ciclesonide being converted at a ratio of 2.5:1 and fluticasone propionate being converted at a ratio of 2:1. Mometasone and beclomethasone were considered equivalent to budesonide.

### 2.4. Outcome

The primary outcome was defined as the finding of *H. influenzae* isolated from a lower respiratory tract sample (i.e., bronchial alveolar lavage, bronchial secretion, tracheal secretion, and sputum samples), determined by a culture within 365 days after cohort entry.

### 2.5. Statistical Analysis

A Cox proportional hazard regression model was used to calculate the risk of acquiring *H. influenzae* associated with ICS use. Death was used as a competing risk in the model since it prevents infection with *H. influenzae*. The analysis was adjusted for the following markers of disease severity and possible confounders: severity of airway obstruction (i.e., percentage of predicted FEV_1_), BMI, smoking status, age, sex, the accumulated dose of oral corticosteroids (OCS) used 365 days prior to cohort entry, and calendar year for entry into the DrCOPD cohort (Figure 2). Unknown FEV_1_ (n = 2429 patients, 11.5%) and BMI measurements (n = 2520 patients, 11.9%) were imputed using the next observation at a following outpatient visit. Unknown smoking status (n = 2322 patients, 10.9%) was categorised as non-active smokers (i.e., most common). Exposure to OCS in the entire prior year was categorised into low and high dose, using the median cumulative dose. Non-users of oral OCS were used as the reference group. ICS was not used as a time-dependent variable because long-term adverse effects of corticosteroids can occur after discontinued use [18,19,20]. 

We also used multivariable-adjusted hazard ratios from corresponding ICS exposure groups to estimate the numbers needed to harm (i.e., the number of individuals treated with ICS that were associated with the finding of one *H. Influenzae* in the cultures) [21].

Continuous variables were reported as median and interquartile values. Group comparisons of continuous variables were performed using parametric and nonparametric tests, where appropriate. Categorical variables were presented as frequencies and proportions. Comparisons between categorical groups were made using Fisher’s Exact Test. 

For adjusted analysis, Cox proportional hazards regression models were used. Statistical significance was defined as a *p* value less than or equal to 0.05. Statistical analyses were performed using SAS statistical software (version 9.4, SAS Institute Inc., Cary, NC, USA).

The models were tested for proportion of hazards and linearity of continuous variables and were found to be valid except for linearity of age. Consequently, age was converted into a categorical variable using the median and quartile values. No interactions were found between accumulated exposure of OCS 365 days prior to cohort entry and accumulated exposure to ICS 365 days prior to cohort entry (*p* value = 0.3) or the type of ICS and accumulated exposure to ICS 365 days prior to cohort entry (*p* value = 0.9). 

### 2.6. Sensitivity Analyses

A propensity score was calculated using the same co-variates used in the main analysis. A Greedy-Match algorithm, created and maintained by biomedical statisticians at the Mayo Clinic [22], was then used to match patients exposed to low and no ICS doses 1:1 with patients exposed to high and moderate doses of ICS. A univariate cox proportional hazard regression model with death as a competing risk was then run to calculate the risk of *H. influenzae* associated with the use of ICS (i.e., high or moderate ICS dose versus low or no ICS dose). A robust variance estimator retested the estimate and showed an absence of independence in the outcome after matching.

## 3. Results

The study cohort included 21,218 patients with COPD. In total, 801 (3.8%) had *H. Influenzae* isolated from a lower respiratory sample within 365 days from cohort entry, with a median time of 147 days (interquartile range (IQR): 60–234 days). Males were 9954 (46.9%) and the rest of the patients, i.e., 11264 (53.1%), were females. All female patients are included in all analyses despite male sex being mentioned. Patients from whom *H. influenzae* was isolated were more likely to have lower BMI, lower FEV_1_, and were more often hospitalized due to exacerbations prior to cohort entry compared to patients from whom *H. influenzae* was not isolated (Table 1). Appendix A lists comorbidities in the study population. Prescriptions of respiratory drugs, oral corticosteroids and antibiotics were more frequently used in the group of patients from whom *H. influenzae* was isolated (Appendix A). The prevalence of ICS use was markedly higher in patients from whom *H. influenzae* was isolated (82%) compared to those without (66%) (Table 2). Moreover, patients from whom *H. influenzae* was isolated were also more likely to be exposed to higher cumulative doses of ICS in the year prior to cohort entry compared to those without (Table 2). Budesonide and fluticasone were the most frequently used ICS types in both groups (Table 2). 

### 3.1. Outcome

Use of ICS was associated with a 118% increased risk of having *H. influenzae* isolated from lower respiratory tract samples compared to non-use in the unadjusted analyses (HR 2.18, 95% CI 1.82–2.62, *p* < 0.0001) (Table 3). This risk remained significant after adjusting for covariates (HR 1.56, 95% CI 1.29–1.89, *p* < 0.0001) (Table 3). There was a strong dose-dependent response relationship with ICS exposure, and up to 90% with high dose ICS use (HR 1.90, 95% CI 1.52–2.38, *p* < 0.0001) (Table 3). Figure 3A shows cumulative incidence curves while Figure 4 shows multivariable-adjusted numbers needed to harm for corresponding ICS exposure groups. We estimated that the number of ICS users required to obtain one positive *H. Influenzae* sample (i.e., number needed to harm) was 60. 

### 3.2. Sensitivity Analyses

A subgroup of 13,324 patients with COPD in the cohort population was formed using a propensity-matched model, consisting of 6662 (50%) patients with high or moderate ICS exposure matched 1:1 with 6662 (50%) patients with low or no ICS exposure (Appendix A). There was a dose-dependent response relationship with ICS exposure with increased ICS dose (low dose HR 1.32, 95% CI 1.00–1.74, *p* = 0.05; moderate dose HR 1.61 95% CI 1.24–2.09, *p* = 0.0004; high dose HR 1.98 95% CI 1.54–2.56, *p* < 0.0001). The cumulative incidence is illustrated in Figure 3B.

## 4. Discussion

In this large, unselected, outpatient cohort, the use of ICS was significantly and independently associated with a substantial and consistent increase in risk of isolating H. influenzae in samples from the lower respiratory tract. Furthermore, the risk increased in a dose-dependent manner. Patients from whom H. influenzae was isolated were more likely to be active smokers, had more advanced GOLD stage, and had relatively lower BMI at entry compared to patients from whom H. influenzae was not isolated. 

Sparse data are currently available on risk factors for acquiring *H. Influenzae* in COPD patients. Several large-population-based studies have shown infectious pulmonary complications of ICS use in COPD patients, such as pneumonia and mycobacterial infections [6,23,24,25,26]. These previous studies are similar to our study, as they are also based on a large number of ICS users [6,24] and since they also show a strong dose-related risk [6,23,24], with patients categorised in a similar manner for ICS exposure, i.e., low, moderate and high [6,24]. However, our data are the first, to the best of our knowledge, that inform on the risk of H. influenzae in relation to ICS. We limited the patient follow-up to a period of 365 days from cohort entry to decrease the likelihood of the outcome occurring from unaccounted factors. In previous studies, it has been suggested that treatment with fluticasone could increase the risk of pneumonia compared to budesonide [10,11]. We could not confirm these findings for *H. influenzae* infection in our study. 

A possible biological mechanism of *H. Influenzae* infection could be ICS-mediated disruption of both innate and adaptive immune responses [27,28], causing an ineffective bacterial clearance and increased bacterial load in the airways [29]. Our data also show an overweight of non-communicable diseases (NCDs) such as asthma, bronchiectasis, diabetes, renal and heart failure in *H. Influenzae*-positive patients (Appendix A). Therefore, it is also important to further investigate the management of these NCDs and their impact on the risk of acquiring H. Influenzae, especially in combination with different therapies for COPD [30].

We believe that our data support the current evidence towards both reducing and stopping maintenance treatment with inhaled corticosteroids in COPD when the patient is at increased risk of future respiratory infections. Unfortunately, there is also a lack of an effective vaccine that prevents against *H. Influenzae* infection because it only covers encapsulated *H. Influenzae* [31], and *H. Influenzae* infection among COPD patients is almost exclusively caused by non-encapsulated *H. Influenzae* [32].

### 4.1. Strengths of Current Study

In our study, the data were obtained from the DrCOPD, which is a nationwide Danish registry with more than 90,000 patients, and with more added each year. The registry is linked digitally with other civil and health registers, which allows for complete follow-up on medication use, mortality, and microbiological data of patients from Eastern Denmark. Therefore, it is not possible to have a positive culture for any micro-organism in Eastern Denmark without this being registered in the database. Moreover, the combined registries contain an abundancy of clinical and demographic parameters needed to adjust for potential confounders, such as the severity of COPD defined by airflow limitation and previous exacerbations, smoking status, BMI, cumulative use of OCS, and prior use of antibiotics. Most importantly, COPD diagnosis has a high accuracy, being spirometry based and specialist verified, and it is validated at least annually. We also tested the statistical model for a possible and relevant interaction between exposure to ICS and accumulated dose of oral OCS prior to analysis (no interactions were found). Moreover, the results were robust to sensitivity analysis with a propensity-matched cohort. We chose not to assess ICS exposure as a time-dependent variable since use of corticosteroids is associated with long-term adverse effects, even after discontinued use [18,19,20]. Furthermore, infectious adverse events have been demonstrated to persist until after 3 months in patients with chronic lung disease [33]. 

We also investigated the number of times microbiological culture specimens were taken from the lower respiratory tract during one year prior to study entry regardless of the organism isolated (Table 1). No apparent difference was detected between *H. Influenzae* -positive and -negative groups. This reduces the possibility of selection bias such that patients from whom *H. influenzae* was isolated delivered samples more often for microbiological workout. 

### 4.2. Limitations of Current Study

As an important limitation to our study, we were not able to observe the actual adherence, and we cannot be certain that all patients used the medication as prescribed. However, lack of adherence would likely tend to underestimate the effect of ICS exposure. Furthermore, most patients repeatedly collected the prescribed medicine, which supports in some way, the notion that they were adherent. Another limitation was our inability to confirm the infection clinically or radiologically in patients from whom *H. influenzae* was isolated from their lower respiratory tract. However, even clinically stable COPD bacterial colonisation causes disease progression [12] and is described by some as a form of chronic infection instead of only colonization [34].

Although our results seem biologically plausible, importantly, a causal relationship cannot be determined due to the observational design of our study. In addition, our results were adjusted for many important confounders, but residual confounding may still be present, and our conclusions should be tested in other large cohorts.

## 5. Conclusions

ICS use in outpatients with COPD is associated with a dose-dependent and substantial increase in risk of acquiring *H. Influenzae*. This observation supports that a thorough clinical assessment of risk of infection should be made before prescription of ICS. We speculate that high-dose ICS should be used with caution, whereas low doses do not seem to carry a similarly high risk of contracting a Haemophilus infection. 

## Figures and Tables

**Figure 1 jcm-11-03539-f001:**
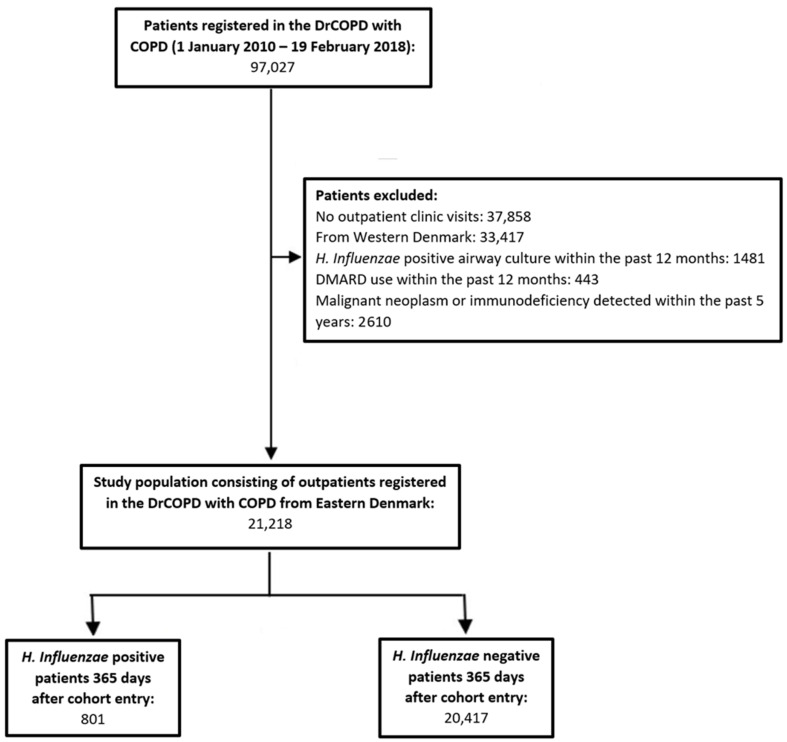
**Study flow chart illustrating the patient selection criteria**. After exclusion, the study population consisted of 21,218 patients, of which 801 acquired H. Influenzae during follow-up of 365 days. Abbreviations: Danish Register of Chronic Obstructive Pulmonary Disease (DrCOPD), disease-modifying anti-rheumatic drugs (DMARD), Haemophilus influenzae (H. Influenzae).

**Figure 2 jcm-11-03539-f002:**
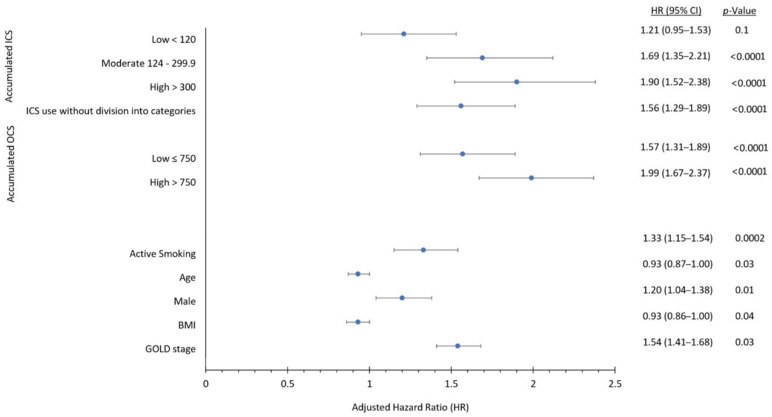
**Forest plot of variables used in Cox proportional hazard regression model showing adjusted hazard ratios with their respective 95% confidence intervals**. The variables include: accumulated ICS dose (low < 120 mg, moderate 120–300 mg, high > 300 mg; reference group; no ICS use), accumulated OCS dose (low ≤ 750 mg; high >750 mg; reference group; no OCS use), active smoking (reference group: non-active smoking), age (per group increase), sex (male; all female patients are included in analysis despite male sex being mentioned), BMI (body mass index; per group increase), and Global Initiative for Chronic Obstructive Lung Disease (GOLD) stage (per group increase).

**Figure 3 jcm-11-03539-f003:**
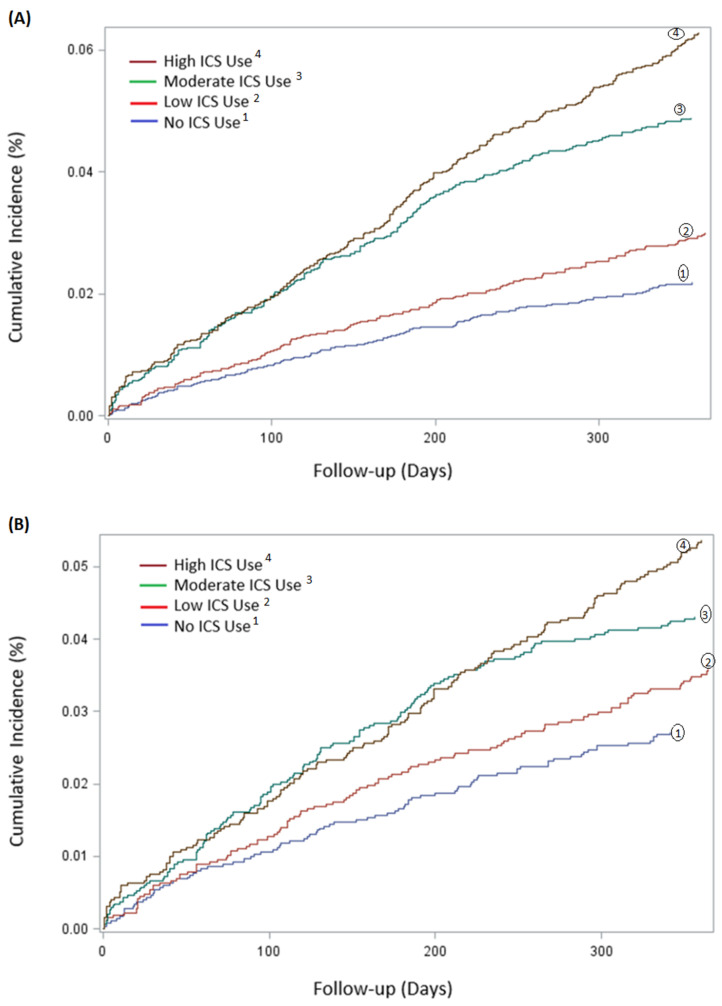
**Cumulative incidence, in percentage (%), of *H. Influenzae* during first 365 follow-up days after cohort entry, categorised according to different accumulated doses of ICS**. (**A**) Cumulative incidence of 21,218 patients, categorised in the following four ICS groups: no ICS use, low ICS dose, moderate ICS dose and high ICS dose. (**B**) Cumulative incidence of 13,324 matched patients, categorized similarly as above.

**Figure 4 jcm-11-03539-f004:**
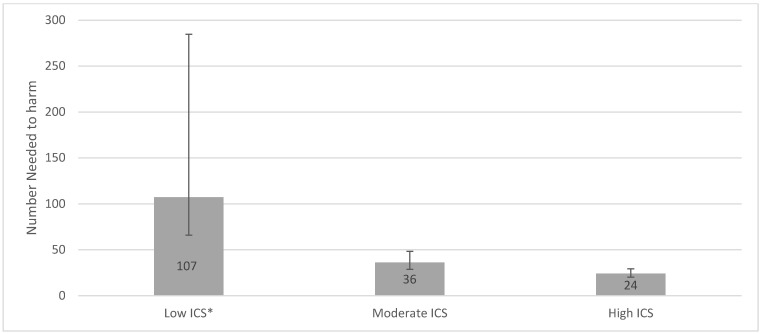
**Number needed to harm for low, moderate, and high ICS use.** ICS exposure was calculated by the accumulated dose of ICS prescriptions reimbursed within 365 days prior to study entry. The accumulated dose was divided into tertiles of low, moderate, and high ICS dose, with non-users as the reference group. The NNH for the three groups: Low ICS (NNH 107, 95% CI 66.1–284.7), Moderate ICS (NNH 36 95% CI 28.8–48.4), and High ICS (NNH 24, 95% CI 20.3–29.4). * There was not a significant increased risk of acquiring a positive *H. Influenzae* sample for the Low ICS group in the main analysis.

**Table 1 jcm-11-03539-t001:** Characteristics at cohort entry in 21,218 patients with COPD.

	All Patients	*H. influenzae* Positive Patients	*H. influenzae* Negative Patients	*p* Value
Number of patients	21,218 (100)	801 (3.8)	20,417 (96.2)	
Demographics				
Age (year), median (IQR)	70 (62–77)	68 (62–75)	70 (62–77)	0.01
Age class				0.0006
Age < 62	5379 (25.4)	206 (25.7)	5173 (25.3)	
Age 62–69	5511 (26.0)	254 (31.7)	5257 (25.8)	
Age 70–77	5632 (26.5)	191 (23.9)	5441 (26.7)	
Age > 77	4696 (22.1)	150 (18.7)	4546 (22.3)	
Male	9954 (46.9)	403 (50.3)	9551 (46.7)	0.05
BMI, median (IQR)	25 (21–29)	24 (20–28)	25 (21–29)	<0.0001
Unknown BMI	2520 (11.9)	49 (6.1)	2471 (12.1)	
BMI class				<0.0001
<18.5	1787 (8.4)	115 (14.4)	1672 (8.2)	
18.5–24.9	7152 (33.7)	293 (36.6)	6859 (33.6)	
25–29.9	5587 (26.3)	211 (26.3)	5376 (26.3)	
30–34.9	2735 (12.9)	90 (11.3)	2645 (13.0)	
≥35	1437 (6.8)	43 (6.1)	1394 (6.8)	
Pulmonary parameters				
MRC, median (IQR)	3 (2–4)	3 (3–4)	3 (2–4)	<0.0001
Unknown MRC	2631 (12.4)	56 (7.0)	2575 (12.6)	
FEV_1_% predicted, median (IQR)	49 (36–63)	38 (29–50)	50 (37–64)	<0.0001
Unknown FEV_1_%	2429 (11.5)	51 (6.4)	2378 (11.7)	
FEV_1_% predicted, severity of obstruction				<0.0001
≥80	1430 (6.01)	19 (2.84)	1411 (6.10)	
79–50	9038 (37.99)	155 (23.13)	8883 (38.42)	
49–30	9868 (41.48)	306 (45.67)	9562 (41.36)	
<30	3453 (14.52)	190 (28.36)	3263 (14.11)	
Smoking status				0.004
Active	7187 (33.9)	326 (40.7)	6861 (33.6)	
Former	11,043 (52.1)	409 (51.1)	10,634 (52.1)	
Never	666 (3.1)	18 (2.3)	648 (3.2)	
Unknown	2322 (10.9)	48 (6.0)	2274 (11.1)	
Hospital requiring COPD exacerbation 12 months prior to cohort entry				<0.0001
0	10,474 (49.4)	309 (38.6)	10,165 (49.8)	
1	2903 (13.7)	130 (16.2)	2773 (13.6)	
≥2	7841 (37.0)	362 (45.2)	7479 (36.6)	
All-cause hospitalisation 12 months prior to cohort entry	18,313 (86.3)	738 (92.1)	17,575 (86.1)	<0.001
Number of pulmonary microbial cultures in the preceding year ^a^	0 (0–2)	1 (0–2)	0 (0–2)	<0.0001

Data are reported as n (%) or median (IQR), unless indicated otherwise. ^a^ Any pulmonary microbiological cultures performed during 365 days prior to cohort entry. Abbreviations: chronic obstructive pulmonary disease (COPD), interquartile range (IQR), Medical Research Council Dyspnoea Scale (MRC), body mass index (BMI) in kg/m^2^, forced expiratory volume (FEV_1_) in the first second.

**Table 2 jcm-11-03539-t002:** Use of ICS 365 days prior to cohort entry in 21,218 patients with COPD. Patients with no ICS use (n = 7076) 365 days prior to cohort entry are not included in the table.

	All Patients	*H. influenzae* Positive Patients	*H. influenzae* Negative Patients	*p* Value
Patients with ICS use	14,142 (66.7)	657 (82.0)	13,485 (66.0)	<0.0001
Accumulated ICS dose in ICS users ^a^				<0.001
Low	4819 (34.1)	143 (21.8)	4676 (34.7)	
Moderate	4559 (32.2)	219 (33.3)	4340 (32.2)	
High	4764 (33.7)	295 (44.9)	4469 (33.1)	
Median accumulated ICS dose, mg (IQR)	202 (96–367)	269 (139–510)	195 (96–360)	
Daily mean ICS dose, ug ^b^	763	984	752	<0.0001
Number of prescriptions, median (IQR)	5 (2–8)	6 (3–9)	5 (2–8)	<0.001
Number of prescriptions by ICS type ^c^				
Beclomethasone	765 (1.0)	99 (2.2)	666 (0.88)	
Budesonid	51,871 (64.6)	2584 (58.4)	49,287 (64.9)	
Fluticasone	27,266 (33.9)	1679 (37.9)	25,587 (33.7)	
Ciclesonide	106 (0.13)	21 (0.47)	84 (0.11)	
Momethasone	322 (0.40)	45 (1.0)	277 (0.36)	

Data are reported as n (%) or median (IQR), unless indicated otherwise. ^a^ Budesonide equivalent doses: low <120 mg; moderate 120–300 mg; high > 300 mg. ^b^ Budesonide equivalent doses. ^c^ Last prescription redeemed prior to cohort entry. Abbreviations: chronic obstructive pulmonary disease (COPD), inhaled corticosteroids (ICS).

**Table 3 jcm-11-03539-t003:** Cox regression hazard estimates for risk of *H. influenzae* with use of inhaled corticosteroids in the study population (n= 21,218) and propensity score-matched population (n = 13,324).

*H. influenzae*	Unadjusted HR (95% CI)	*p* Value	Adjusted HR (95% CI)	*p* Value	HR after Matching (95% CI)	*p* Value
Accumulated ICS dose 365 days prior cohort entry, mg ^a^						
Low <120	1.37 (1.21–1.73)	0.008	1.21 (0.95–1.53)	0.1	1.32 (1.00–1.74)	0.05
Moderate 120–300	2.27 (1.84–2.81)	<0.0001	1.69 (1.35–2.12)	<0.0001	1.61 (1.24–2.09)	0.0004
High > 300	2.92 (2.39–3.57)	<0.0001	1.90 (1.52–2.38)	<0.0001	1.98 (1.54–2.56)	<0.0001
ICS use without division into categories	2.18 (1.82–2.62)	<0.0001	1.56 (1.29–1.89)	<0.0001		
Accumulated OCS dose 365 days prior to cohort entry, mg ^b^						
Low ≤750	1.89 (1.58–2.26)	<0.0001	1.57 (1.31–1.89)	<0.0001		
High >750	2.67 (2.28–3.14)	<0.0001	1.99 (1.67–2.37)	<0.0001		
Active smoking ^c^	1.32 (1.14–1.53)	0.0003	1.33 (1.15–1.54)	0.0002		
Age class						
Age <62	Reference		Reference			
Age 62–69	1.23 (1.02–1.48)	0.03	1.11(0.92–1.34)	0.3		
Age 70–77	0.89(0.73–1.08)	0.2	0.83 (0.68–1.02)	0.08		
Age >77	0.84 (0.68–1.04)	0.1	0.84 (0.67–1.05)	0.1		
Male	1.17 (1.02–1.35)	0.02	1.20 (1.04–1.38)	0.01		
BMI class ^d^						
<18.5	1.84 (1.49–2.28)	<0.0001	1.38 (1.10–1.73)	0.005		
18.5–24.9	Reference		Reference			
25–29.9	1.07 (0.9–1.3)	0.4	1.04 (0.88–1.24)	0.6		
30–34.9	0.94 (0.75–1.19)	0.6	0.93 (0.74–1.18)	0.5		
≥35	0.87 (0.63–1.19)	0.4	0.86 (0.62–1.18)	0.3		
GOLD stage ^e^						
Stage 1: FEV_1_% ≥ 80	Reference		Reference			
Stage 2: FEV_1_%= 79–50	1.38(0.88–2.16)	0.2	1.21 (0.77–1.90)	0.4		
Stage 3: FEV_1_% = 49–30	3.22(2.07–5.01)	<0.0001	2.29 (1.46–3.58)	0.0003		
Stage 4: FEV_1_% < 30	5.05 (3.22–7.90)	<0.0001	2.81 (1.76–4.48)	<0.0001		

Abbreviations: inhaled corticosteroids (ICS), body mass index (BMI) in kg/m^2^, oral corticosteroids (OCS), forced expiratory volume (FEV_1_) in the first second. The model is adjusted for calendar year for study entry and all variables displayed in the table. ^a^ Reference: no ICS use. ^b^ Reference: no OCS use. ^c^ Reference: never or former smoking. ^d^ BMI class (kg/m^2^). ^e^ Increase in predicted FEV_1_% severity stage defined by the Global Initiative for Chronic Obstructive Lung Disease (GOLD).

## Data Availability

Due to the Danish legislation regarding sharing of population data, source data collected for this study cannot be made available for others. However, Danish citizens who have a legitimate reason can apply for access to the data via the Danish National Health Authority (https://sundhedsdatastyrelsen.dk. Last accessed on 17 June 2022).

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
