# Peer review of "Use of Inhaled Corticosteroids and Risk of Acquiring *Haemophilus influenzae* in Patients with Chronic Obstructive Pulmonary Disease"

_jcm, 2022, doi:10.3390/jcm11123539_

Round 1

Reviewer 1 Report

An interesting study conducted with proper methodology by linking several high quality administrative registers in one area of ​​Denmark. A significant association has been demonstrated between ICS dose dependent exposure and the incidence of Haemophilus Influentiae infection in COPD patients. The limitations of an observational study with respect to highlighting a causal relationship beyond mere association are clearly set out in the discussion.In general, I agree on the need, further underlined by the results of this study, to use ICS in COPD patients in the light of an explicit benefit / risk assessment. 

However, it seems to me that the potential benefits of a therapy conducted with ICS and LABA / LAMA are definitely underestimated in the introduction of the manuscript where it is read "Moreover, multiple studies show none, or minimal improvement in survival benefits associated with 52 ICS use in COPD".  Recently published data from IMPACT and ETHOS study (the largest studies available on this topic) show an important reduction of the absolute risk of  all cause mortality by adding ICS to LABA/LAMA. I suggest using a more balanced approach to describe the potential benefits of ICS therapy when presenting the results of a study highlighting a potential and important risk associated with the same therapy. 

Author Response

An interesting study conducted with proper methodology by linking several high quality administrative registers in one area of ​​Denmark. A significant association has been demonstrated between ICS dose dependent exposure and the incidence of Haemophilus Influentiae infection in COPD patients. The limitations of an observational study with respect to highlighting a causal relationship beyond mere association are clearly set out in the discussion.In general, I agree on the need, further underlined by the results of this study, to use ICS in COPD patients in the light of an explicit benefit / risk assessment. 

It seems to me that the potential benefits of a therapy conducted with ICS and LABA / LAMA are definitely underestimated in the introduction of the manuscript where it is read "Moreover, multiple studies show none, or minimal improvement in survival benefits associated with 52 ICS use in COPD". Recently published data from IMPACT and ETHOS study (the largest studies available on this topic) show an important reduction of the absolute risk of all-cause mortality by adding ICS to LABA/LAMA. I suggest using a more balanced approach to describe the potential benefits of ICS therapy when presenting the results of a study highlighting a potential and important risk associated with the same therapy.

R_C1: Thank you for this comment. It is correct that IMPACT and ETHOS show a reduction in mortality. However, patients were highly selected (patients chosen had low risk of pneumonia, no history of earlier infections, no bronchiectasis or immunomodulating therapies, please see supplement to the protocol). Moreover, in IMPACT there was an over representation of patients with an asthma-like profile and abrupt discontinuation of ICS (see reference number 7 mentioned below).
Furthermore, in ETHOS a dose-response effect with regards to mortality could not be shown (reduction in mortality was present only with high dose budesonide)8. Thus, the conclusion of ETHOS and IMPACT with regards to mortality is not without controversy. This is updated in the. article
.
Page 3 line 7 – 13: “ It is true that ETHOS and IMPACT show a reduction in all-cause mortality with ICS use, though this conclusion is considered inaccurate by some7,8.Moreover, a network metanalysis of ETHOS, KRONOS, IMPACT, and TRILOGY studies also show an increased risk of pneumonia with no improvement in mortality despite decreased risk of acute exacerbation of COPD with use of ICS in dual or triple fixed dose containers8. Eventually, some other studies have also
shown none 9,10 or minimal 11 improvement in survival benefits associated with ICS use in COPD.”
New reference (#7): “Suissa, S., Mortality in IMPACT: Confounded by Asthma? Am J Respir Crit Care Med 2020, 202 (5), 772-773.”

Reviewer 2 Report

1.       COPD is a non-communicable disease, sometimes associated with other communicable lung infections such as tuberculosis, where corticosteroids may be used as host-directed therapies to treat the disease. Please have a look at this article and discussion about this in the discussion. This is important.

https://pubmed.ncbi.nlm.nih.gov/30876870/

2.       Did the authors check the studied patient’s history for other associated lung diseases?

3.       Only 3.8% of patients got infected with H. influenzae, how do authors confirm that this is only because of ICS? There may be many other factors for this infection. Please discuss and justify.

4.       What is the number of male and female patients included in the study? Please include the details.

5.       Figure2. Please include the statistical significance details in the figure or figure legends. Dose only males included in this study? What about the controls here? Please include the details in the figure legend.

6.       Figure3. What is the difference between both panels? It is suggested that this data can be better represented with a bar graph along with statistical significance between different groups.

7.       Figure4. What type of number about, the authors mentioned here? Please put all the details in the figure legend and the figure as well. What about the controls here? Please include the statistical significance in the figure and details in the figure legend too.

8.       This study has a lot of limitations. Please include a separate section entitled “Limitation of the current study”.

Author Response

C1: COPD is a non-communicable disease, sometimes associated with other communicable lung infections such as tuberculosis, where corticosteroids may be used as host-directed therapies to treat the disease. Please have a look at this article and discussion about this in the discussion. This is important. Https://pubmed.ncbi.nlm.nih.gov/30876870/

C2:       Did the authors check the studied patient’s history for other associated lung diseases?

R_C1&2: Thank you for these comments and sharing a very thought-provoking article.

The above-mentioned article does mention a beneficial role of corticosteroid on TB meningitis and pericarditis but has no concrete opinion on use of corticosteroids on pulmonary tuberculosis.

In our study we have looked at other comorbidities including lung diseases such as asthma and bronchiectasis and apparently there seems to be an overweight of these comorbidities in H.Influenza positive group. And we think in this regard it is very important to think of ICS as a host-directed therapy which can influence pathophysiology of both the infection and other co-occurring non communicable diseases.

Following has been added to the discussion:

Page 12 line 21-25: “Our data has also shown an overweight of non-communicable diseases (NCDs) such as asthma, bronchiectasis, diabetes, renal and heart failure in H. Influenzae positive patients (Supplemental Table 2). Therefore, it is also important to further investigate the management of these NCDs, and their impact on the risk of acquiring H. Influenzae, especially in combination with different therapies for COPD 30.”

New reference #30: “Baindara, P., Host-directed therapies to combat tuberculosis and associated non-communicable diseases. Microb Pathog 2019, 130, 156-168”

C3: Table 2: Only 3.8% of patients got infected with H. influenzae, how do authors confirm that this is only because of ICS? There may be many other factors for this infection. Please discuss and justify.

R_C3:  Thank you for this comment. In limitations of this study we discus that we cannot establish a causal relationship between ICS use and H. influenzae infection, what we show is just an association,  and to strengthen this association we have used certain exclusion criteria such as no H. influenzae infection and no use of DMARDs in previous 1 year, no patients with immunodeficiencies and or with malignant neoplasm in previous 5 years. Moreover we adjusted our hazard regression model for other potential cofounder such as severity of airway obstruction (i.e. percentage of predicted FEV1), BMI, smoking status, age, sex, the accumulated dose of oral corticosteroids (OCS) used 365 days prior to cohort entry, and calendar year for entry in DrCOPD-cohort. We also performed a sensitivity analysis by propensity matching the patients to decrease the chances of confounding.

C4: What is the number of male and female patients included in the study? Please include the details.

R_C4: Male   patients are 9,954 (46.9%) and the rest of patients i.e. 11264 (53.1%) are females. We have chosen male as a proxy for both sexes in analyses. All female patients are included despite male sex being mentioned.

Page 11 line 4-5: “Males were 9,954 (46.9%) and rest of the patients i.e. 11264 (53.1%) were females. All female patients are included in all analyses despite male sex being mentioned.

C5: Figure2. Please include the statistical significance details in the figure or figure legends. Dose only males included in this study. What about the controls here? Please include the details in the figure legend.

R_C5: Thank you for the comment we have added statistical significance in figure 2. As mentioned above we use male gender as proxy for both sexes. Thus, both male and females are included in the study. Controls are mentioned as reference group in figure legend already.

Unfortunately figure 2 got mixed with an earlier version while submitting the manuscript. This has also been updated according to our latest data.

Page 8 line 1:

Page 8 line 6-7: “sex (male; all female patients are included in analysis despite male sex being mentioned)”.

C6: Figure3. What is the difference between both panels? It is suggested that this data can be better represented with a bar graph along with statistical significance between different groups.

R_C6: Figure 3 panel A shows cumulative incidence in complete/unselected population and figure 3 panel B shows cumulative incidence in a smaller group of matched population.

As we have shown cumulative incidence as a function of time i.e. follow up time it can only be shown with a line graph and it is not possible to use a bar graph.

C7: Figure4. What type of number about, the authors mentioned here? Please put all the details in the figure legend and the figure as well. What about the controls here? Please include the statistical significance in the figure and details in the figure legend too.

R_C7: Thank you for this comment. Figure 4 shows number needed to harm. The NNH is independent of the P-value as it shows the clinical significance (vs statistical significance). We have though added confidence intervals to the figure and legend. Non-users of ICS are used as a reference group (control group).

Page 10 line 1:

Page 10 line 5-6: “The NNH for the three groups: Low ICS (NNH 107, 95% CI 66.1-284.7), Moderate ICS (NNH 36 95% CI 28.8-48.4), and High ICS (NNH 24, 95% CI 20.3-29.4).”

C8: This study has a lot of limitations. Please include a separate section entitled “Limitation of the current study”.

R_C8: Thank you for this comment. We have added a separate section for both limitations and strengths.

Page 13 line 1: “Strengths of current study”

Page 13 line 22: “Limitations of current study

Round 2

Reviewer 2 Report

The authors successfully responded and updated the manuscript as per the reviewer's suggestions/comments.